# Feminist Academic Activism in English Language Teaching: The Need to Study Discourses on Femininities Critically

Esteban Francisco López-Medina

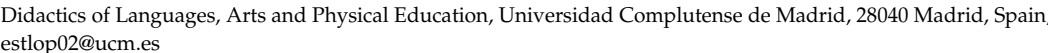

Didactics of Languages, Arts and Physical Education, Universidad Complutense de Madrid, 28040 Madrid, Spain; estlop02@ucm.es

**Abstract:** Social research into English Language Teaching (ELT) has a long history. Within it, gender studies have gained ground in recent decades, with special focus on materials and resources. However, a proper integration of the category of femininity has not yet been achieved. The article offers an ample, argumentative, narrative literature review of the main realizations of femininity, as theorized in recent years, such as emphasized femininity or entitled femininity, as well as some other concepts like ambivalent sexism and postfeminism. It is written as the second item within a series of papers that aims to theoretically support the assumption of Feminist Critical Discourse Analysis as a suitable method to discover how all these social phenomena interact in ELT contexts, helping to shape its (gender's) hidden curriculum. The paper concludes the necessity of integrating the issue of femininities in teacher training programs and in social research in ELT, for the sake of making this field more a liberating practice and less a means of (re)production of (gender) inequalities. To do so, it offers areas of interest for critical researchers and ELT practitioners to carry out such empirical investigation, which is the upcoming stage in this sequence of publications.

**Keywords:** English Language Teaching; femininity; postfeminism; research; sexism

## 1. Introduction

In past works [1], I have advocated for the assumption of the concept of "hegemonic masculinity" to analyze teaching practices and resources in the teaching of English as "an other" tongue: "built over [ . . . ] difference rather than on national and imperial territories" [2] (p. 249), that is, teaching awareness of the inequalities enacted in the relationship between languages. Moreover, it is often argued that binary conceptions around male and female should be debunked [3]. However, given that Western societies are primarily built upon such constructs, it makes sense to also suggest the use of "femininity" to obtain further insight into social phenomena in general and the world of English language teaching (ELT) in particular, which was the primary objective of this work.

However, when delving into research and reading about "emphasized femininity" [4]—the concept originally aimed at in this text—the scope needs to change. One publication led to the other, and emphasized femininity alone was revealed to be insufficient to describe the wide array of ideas around women or their interaction in the field of ELT. Based on the social character of gender [5], this article aims to theoretically argue the possibility of unveiling the intricate workings of social constructs such as emphasized femininity [4], ambivalent sexism [6–8], and postfeminism [9], through feminist critical discourse analysis [10,11].

This article, then, constitutes an instance of an argumentative, narrative literature review. This kind of research intends to present the current works around a topic, without necessarily carrying out a comprehensive search technique or a systematic selection process for the primary sources [12], focusing mainly on recent, readily available literature [13]. Still, while reading the text, the reader will notice that the works of Connell [4], Martin [5], Connor, Glick, and Fiske [6], Glick and Fiske [8], and Lazar [9–11] are fundamental.

After presenting these authors' concepts, the text concludes the need to integrate these meaningful variables into ELT training, teaching practices, and research, offering specific areas of application. In short, the article serves as theoretical support for upcoming empirical research that will apply such proposals in the field of ELT.

## 2. Construction of Gendered Identities (Also in ELT)

The history of the concept of gender is long and rich. Though at the beginning it tended to be opposed to physical sex [14,15], the complex relationship between sociality and physicality soon became clear [16]. Even more, in poststructuralist times, queer authors such as Butler [17] have problematized the traditional clear-cut differentiation between physical sex and cultural gender, advocating for a holistic experience of both.

The core social aspect of gender—and sex—cannot be denied, which is why it should be understood as a social institution, implying that it tends to persist and interact with other social institutions [5]. Connell [4] helps us understand this when she urges the abandonment of individualistically and psychologically reductionistic stances on gender to, instead, embrace its collective, historical, material, practical, and institutional properties.

> In common-sense understanding gender is a property of individual people. When biological determinism is abandoned, gender is still seen in terms of socially produced individual character. It is a considerable leap to think of gender as being also a property of collectivities, institutions, and historical processes. [4] (p. 139)

Connell [4] also explains how the socially embodied practices of gender help to configure the structures of gender, resulting in a cyclical practice that continuously transforms it. As a result, gender becomes a social institution which remains, underlying apparent changes. She also remarks how gender, as a social institution that permeates society and individuals, is constantly adapting and transforming to persist, which entails it to be historical but also powerful, enabling it to accomplish things.

Understanding gender as a social institution paves the way to the realization of its sociality and agency, undermining the widespread assumptions that it is essentially, biologically, or naturally determined by individual bodies [3]. As Martin puts it, "gender 'does things' with and to bodies but gender is not explainable by or reducible to the body" [5] (p. 1260), thus explaining what Butler [17] labels as the performative character of gender.

If gender is a social institution, that is, part of social organization, as there will always be competing interests in any social group, attention must be paid to how power is enacted through it [18]. In fact, Hill Collins [19] lists gender, race/ethnicity, sexuality, social class, and age as the "axes of difference"; that is, social institutions that structure, distribute, and symbolize power, both per se and intersectionally.

"The borrowing of gendered expectations to create and legitimate social relations in all or most other institutions is a clear indicator of its institutional power" [5] (p. 1266). Even if neither gender nor other social institutions can be regarded as more foundational than one another, it is important to conceptualize it as such to contribute to further insight into social phenomena such as, in our case, the teaching of languages and, more specifically, the teaching of English.

> When one uses English to ask for a bottle of wine in Italy, the intention is to obtain a bottle of wine. But in speaking English instead of Italian, speakers—in practice—contribute to the hegemony of English worldwide, irrespective of their intentions. Similarly, people who practice gender at work without intending to can and do produce harm. [5] (pp. 1262–1263)

This quote can easily reveal what happens in the teaching of "other tongues" [2], specifically in ELT: practitioners may not only reinforce the hegemony of the language they teach, but, unconsciously, they can also produce and reproduce the social constraints of gender, which is why it is worth researching teaching contexts. As Gramsci [20] well

explains, schools tend to reproduce hegemonic ideologies through their formal and hidden curricula, gender and patriarchy being among them.

In light of the above, the social institution of gender can be defined as "a set of repeated acts within a highly rigid regulatory frame that congeal over time to produce the appearance of substance of a natural sort of being" [17] (p. 45). In short, gender must be understood as a relational social institution which is built through a game of "historically specific organisations of language" [17] (p. 145) or, in other words, discourses. These can be described as "competing ways of giving meaning to the world and of organizing social institutions and processes [ . . . ] [offering] the individual a range of modes of subjectivity" [21] (p. 35). This is precisely what happens in classrooms in general and in ELT contexts in particular: they cannot help conveying a wide array of discourses beyond language itself, gender being one of the most prominent. These "other" non-intended discourses that can be found in educational institutions have often been labeled as a "hidden curriculum" [22].

Ample research in recent decades has demonstrated that teaching materials and practices in ELT imply, like any other teaching, a hidden curriculum. The one studied in ELT, despite discreet improvements, still tends to perpetuate anachronistic gender roles constructed in a discursive way within a binary, relational, dualistic framework [23–31], "in which the subordinated term is negated, rather than the two sides being in equal balance" [32] (p. 256). The result is a discourse which presents femininity as "a variety of negations of the masculine" [32] (p. 256), enacting a hierarchy of power which places masculinities at the top and femininities at the bottom [4,33].

Davies [34] states that no "official" message such as this is automatically assimilated by students, as each of them with agency negotiates discourses in a subjective way. In consequence, the final meaning learners will be assigned to masculinities and femininities, which will depend on a lot of variables and cannot be presumed based on just educational analysis alone. Still, it is worth reviewing a series of concepts around femininities and "doing girl and woman" [32,35] that typically populate classrooms and should be taken into account when studying teaching realities.

### 3. Towards Academic Activism: Feminist Critical Discourse Analysis

The previous section evidenced the social construction of a binary gendered system of power around men and women and around masculinity and femininity. Such concepts, characteristically relational, oppositional, and situational, cannot be reduced to a simple definition. Instead, they must be understood as "cluster concepts":

> . . . not amenable to straightforward definition but [ . . . ] recognized through a cluster of attributes, some of which are more salient than others, but which may not all be present. The gender binary, in consequence, only operates at the level of the label. There are only two labels, but what they denote will vary considerably between situations, and will frequently overlap. [32] (pp. 258–259)

If these social institutions of femininity and masculinity are discursively constructed and negotiated in all spheres of life, educational contexts included, we need an apt method to help us uncover what their "cluster features" are. We then choose Feminist Critical Discourse Analysis, as posed by Michelle Lazar:

> The aim of feminist critical discourse studies, therefore, is to show up the complex, subtle, and sometimes not so subtle, ways in which frequently taken-for-granted gendered assumptions and hegemonic power relations are discursively produced, sustained, negotiated, and challenged in different contexts and communities. Such an interest is not merely an academic de-construction of texts and talk for its own sake, but comes from an acknowledgement that the issues dealt with (in view of effecting social change) have material and phenomenological consequences for groups of women and men in specific communities. [11] (p. 142)

Critical Discourse Analysis is based on the understanding of the dialectical relationship of social practices and discourse, which represent and constitute each other [36]. On top of that, Feminist Critical Discourse Analysis helps reveal that the enactment of power in such social practices and discourses is far from neutral but deeply and unfairly gendered, favoring men over women, thus putting dominating masculinities and subordinated femininities in practice. As Feminist Critical Discourse Analysis aims to uncover and subvert such dynamics, Lazar [11] also refers to it as "analytical activism" or "academic activism". Because gender is conceptualized as an unfair ideological structure, Feminist Critical Discourse Analysis, "raising critical awareness through research and teaching [ . . . ], cannot and does not pretend to adopt a neutral stance; it is scholarship that makes its [feminist] biases part of its argument" [11] (p. 146).

Feminist Critical Discourse Analysis, applied to written and spoken discourses in education in general and ELT in particular, seems to be the most appropriate tool to unearth the dynamics of masculinities and femininities in teaching–learning contexts, as well as to identify their "cluster features". The following sections will focus on discourses around femininities that seem relevant in the Western world in the first third of the 21st century (describing masculinities would exceed the scope of the article. Still, given the relational connection between masculinities and femininities, I have dealt with the former's role in ELT in this article: [1]).

## 4. Femininity for the Sake of Masculine Power

Mimi Schippers claimed in 2007 that "femininity is still decidedly under-theorized" [37] (p. 85), and this probably remains true. A good starting point is Iris Marion Young, who, accordingly with Simone de Beauvoir, took it:

> to designate not a mysterious quality or essence which all women have by virtue of their being biologically female. It is, rather, a set of structures and conditions which delimit the typical *situation* of being a woman in a particular society, as well as the typical way in which this situation is lived by the women themselves. [ . . . ] The female person who enacts the existence of women in patriarchal society must therefore live a contradiction: as human she is a free subject who participates in transcendence, but her situation as a woman denies her that subjectivity and transcendence. [38] (pp. 140–141, emphasis in the original)

It seems clear, then, that femininity is a situational concept that cannot be separated from the experience of women in relation to men. The most popular concept in this field is Raewyn Connell's "emphasized femininity" [4,39]. Though one could be tempted to understand it as a mirror concept of what she labeled as "hegemonic masculinity", the author herself warns against it.

In fact, her notion of hegemonic masculinity has been widely criticized [40–42], and she has revisited her own work with James W. Messerschmidt [39]. Their reflections around Connell's original ideas on hegemonic masculinity can and should be considered when approaching emphasized femininity.

First of all, it is of utmost importance not to univocally associate masculinity with men and femininity with women because, the same way women may "appropriate aspects of hegemonic masculinity" [39] (p. 847), men can also appropriate aspects of femininity. The urge not to reduce hegemonic masculinity to a fixed cluster of features should also be applied to emphasized femininity: if masculinity "represents not a certain type of man but, rather, a way that men position themselves through discursive practices" [39] (p. 841), femininity cannot be otherwise. Finally, they identify some core features of the original definition of hegemonic masculinity that should be kept. Conveniently adapted, they also apply to femininity and can be explained this way: the existence of diverse femininities, the "desirability" of emphasized femininity among them, and their adaptability in order to promote the patriarchal order.

In addition, when describing the variety of manifestations of femininities, Connell also provides a definition of emphasized femininity:

> One form [of femininity] is defined around compliance with this subordination and is oriented to accommodating the interests and desires of men. I will call this "emphasized femininity". Others are defined centrally by strategies of resistance or forms of non-compliance. Others again are defined by complex strategic combinations of compliance, resistance and co-operation. [4] (pp. 184–185)

If, from a conceptual point of view, there are so many similarities when describing masculinities and femininities, why does Connell not use the label "hegemonic femininity"? She clearly states that "all forms of femininity in this society are constructed in the context of the overall subordination of women to men. For this reason, there is no femininity that holds among women the position held by hegemonic masculinity among men" [4] (p.187).

Even the manifestation of femininity that seems to be "privileged" and an object of desire performs the only function of reinforcing patriarchy, the domination of some men and the overall subordination of women. It then becomes clear that speaking of "hegemonic femininity" would be totally misleading and counterproductive as an analytical label to study and explain gendered social phenomena.

Connell's work around emphasized femininity, though path-breaking, has not deepened as much into it as she has in the case of hegemonic masculinity. Sometimes, it has been understood as some sort of hyperfemininity, as a counterpart of hypermasculinity, particularly in the field of psychology:

> An exaggerated adherence to the stereotypic feminine gender role, involving the use of sexuality to gain or maintain romantic relationships with men, the belief that these romantic relationships define their success, and the preference for traditional male behaviors in their partners. [43] (p. 479)

Other authors, such as Schippers, have suggested an alternative model based upon Connell's original proposal and prefer the label "hegemonic femininity" to describe an "idealized quality content [which] consists of the characteristics defined as womanly that establish and legitimate a hierarchical and complementary relationship to hegemonic masculinity and that, by doing so, guarantee the dominant position of men and the subordination of women" [37] (p. 94). This hegemonic manifestation of femininity, the author posits, is in a better position compared to "pariah femininities", "deemed, not so much inferior, as contaminating to the relationship between masculinity and femininity" [37] (p. 95). The author lists some examples of characteristics or practices that, despite being nuclear to the performance of hegemonic masculinity by men, end up stigmatized when actively enacted by women: "desire for the feminine object (lesbian), authority (bitch), being physically violent ('badass' girl), taking charge and not being compliant (bitch, but also 'cock-teaser' and slut)" [37] (p. 95).

Beyond labels, research around femininity and its "privileged" forms seems unanimous in its essential role in perpetuating male domination, commonly known as patriarchy or, sometimes, sexism. In this interaction between femininity and masculinity, it is also useful to resort to the theory of ambivalent sexism [6–8]. Inspired by the theories around ambivalent racism, the authors "realized that gender relations and, therefore, sexist attitudes, differ from race relations because men and women so often lead intimately intertwined lives, whereas Blacks and Whites typically experience much less contact" [8] (p. 530). As a result, sexism reveals itself to be ambivalent, comprising both hostile and benevolent attitudes which, far from conflicting, are, in fact, complementary. This implies that one does not suppress the other; contrarily, they can and do effectively co-exist.

> The very same men who said "can't live with them" were the ones who also said "can't live without them." [ ... ] HS [Hostile Sexism] and BS [Benevolent sexism] were two sides of a sexist coin. And this double-sided coin (even if its hostile component had become more subtle) was at least as ancient as polarized stereotypes of the Madonna and Mary Magdalene. BS was the carrot aimed at enticing women to enact traditional roles and HS was the stick used to punish them when they resisted. [8] (p. 532)



The authors identify three main areas in which sexism can manifest both benevolently and in a hostile way: power, gender differentiation, and sexuality, as shown in Table 1:

**Table 1.** Components of benevolent and hostile sexism.

| Area | Benevolent Sexism | Hostile Sexism |
| --- | --- | --- |
| Power | Protective paternalism | Dominative paternalism |
| Gender differentiation | Idealization of women | Derogatory beliefs |
| Sexuality | Desire for intimate relations | Heterosexual hostility |

Source: Based on [8].

Regardless of their benevolent or hostile nature, all of these attitudes serve to perpetuate traditional gender roles and reinforce the patriarchal social order. Their main difference relies in the subjectively positive or negative meaning they have for the sexist bearer. This means that the benevolent aspects, though subtler and apparently "positive", have a role when engaging women in the workings of the patriarchal system. Similarly, the hostile attitudes aim at preventing them from escaping from it, thus enacting what Jackman [44] labeled as "the iron fist within the velvet glove".

> By offering male protection and provision to women in exchange for their compliance, benevolent sexism recruits women as unwitting participants in their own subjugation, thereby obviating overt coercion. Hostile sexism serves to safeguard the status quo by punishing those who deviate from traditional gender roles. [6] (p. 295)

Feminist research in the last century has helped to uncover the intricate relationships between masculinities, femininities, sexism, and patriarchy which, if sometimes subtler, are in no way less effective. However, in recent decades some postfeminist theorists have suggested that the contradiction between femininity and equality has been effectively overcome, a balance between them thus being possible.

## 5. Entitled Femininity: A Postfeminist Identity Discourse for a Post-Critique Time

In a previous section, the value of Feminist Critical Discourse Analysis to uncover the workings of patriarchy was posed. The main supporter of this stance, Michelle Lazar, makes it clear that the current global neoliberal discourse of postfeminism is one of the most important to address.

> According to this discourse, once certain equality indicators (such as rights to educational access, labour force participation, property ownership, and abortion and fertility) have been achieved by women, feminism is considered to have outlived its purpose and ceases to be of relevance. [11] (p. 154)

This powerful discourse, which started in the USA and the UK in the 1980s and became evident in the 1990s, has permeated numerous social sectors and effectively neutralizes the social struggle intrinsic to feminism, which has for long claimed that "the personal is political" [45]. In short, postfeminism suggests that, once the legal impediments have decayed, it is just a personal issue for each woman to push strongly enough to achieve equal opportunities and equal rights, thus blurring the several constraints many groups of women face to do so and strengthening a "de-politicized ethos" built around a self-centered "me-feminism" rather than the collective "we-feminism" [9,11].

> By and large, postfeminism speaks the language of feminism, but without investment in feminist activism, collectivism, social justice and transformation of prevailing gender orders. [ . . . ] In fact, postfeminism quite typically encompasses a variety of contradictory positions with respect to feminism (Projansky, 2001) all at once, such as pro-feminist and celebratory; anti-feminist backlash; and non-feminist in embracing normative patriarchal practices. [9] (p. 340)

Lazar [46] identifies "entitled femininity" as one of the most significant manifestations of postfeminism, spread by the mass and social media, particularly the advertising industry.

The author points out three different themes that constitute the postfeminist entitled feminine identity. Contrarily to traditional assumptions of women being centered on others—primarily husband and children—entitled femininity "is an entitlement to live a self-absorbed, hedonistic and narcissistic lifestyle based upon consumerist values" [46] (p. 375). The second main constituent of entitled femininity is the "celebration of femininity" by reclaiming and positively re-signifying traditional feminine stereotypes. If the second-wave feminism opposed feminine and feminist, postfeminism undoes their contradiction to claim normative feminine stereotypes as beneficial rather than detrimental, even liberating and emancipating, somehow promoting a "feminist hyperfemininity". Finally, the third essential component of entitled femininity is the celebration of girlhood: "instant self-gratification and pleasure, but also specifically emphasizes youthfulness as a time of fun" [46] (p. 390).

The description of these three core components of entitled femininity helps us become aware of the fact that it is an identity discourse built around self-pleasure and the individual right to consume. As a result, the endorsement of these ideas leads to a non-critical vital attitude which accomplishes nothing but to strengthen the perpetuation of differences and inequalities. Probably, the greatest evidence for this is the fact that the ideal postfeminist woman already described cannot be achieved by all women unless they meet certain age, physical, or wealth requirements, among others. Therefore, the assumption of postfeminist ideas promotes a post-critique stance in life which ends up disempowering the fight for social justice in general and for women's rights in particular.

> The emphasis on fun and pleasure-seeking, she argues, numbs resistance and critique. Yet, the feminine subject, based on an entitlement to consume, is a very particular kind (middle class, heterosexual and willing to "do" youthfulness), which although it appears pro-women, excludes many women and creates inequalities among them. [46] (p. 342)

It seems clear that, when seen critically, the postfeminist discourse is radically problematic. As a matter of fact, it would not be going too far to label it as a post-critique 21st-century "feminine mystique", subtler than the one exposed by Betty Friedan [47] in the 20th century, but equally—if not more—effective when reinforcing the patriarchal order.

## 6. Discussion

Throughout the last pages, the most relevant ideas around femininity have been presented: gender, emphasized femininity, ambivalent sexism, postfeminism, and entitled femininity among others. Feminist Critical Discourse Analysis [10,11] has been suggested as a proper way to discover how all of them—together with masculinities—interact in educational contexts in general, with a special focus on the teaching of English, understood as another tongue [2].

Since the late 20th century, the school has been reconceptualized as more than just a place for the acquisition of knowledge but a context to get ready for life, to learn to know, to do, to live with others, and to be [48]. What is more, the European Union has recommended the inclusion of key competences for lifelong learning as essential components of the education system in each state member [49], which include not only linguistic or scientific mentions but also sociality and citizenship, with explicit reference to gender equality. Given the fact that all academic subjects should contribute to the development of all these competences, it then becomes clear that the teaching of languages cannot be an exception when it comes to it.

One of the aims when developing the social and citizenship competences should be to "free ourselves, both as researchers and as individuals, from binary conceptions of masculinity and femininity that constrain both what we can think and who we can be" [32] (p. 262). These binary conceptions populate ELT environments [23–31], which have their fair share of responsibility for the perpetuation of unequal binary gendered societies. These authors have described a wide array of instances of how they do so through, for example, stereotypical representations of women and men, the silencing of minoritized identities like LGBTIQ+ people, and speakers of other languages or non-white individuals, together with

the recurrent bias towards the pre-eminence of the high/middle-class, white, heterosexual, male "native-speaker" from Western countries like the USA and the UK.

Contrarily, we pose that ELT contexts, like any other teaching practices, ought to be part of "a degendering movement whose goal is greater equality [and] would also have to include pressure for erasure of other invidious divisions, especially race and ethnicity, and for open access to economic resources, educational opportunities and political power" [3] (p. 90).

The first step towards this aim is, undoubtedly, the acknowledgement of the role of the education system in the building of this unequal binary gendered society through the enactment of masculinities and femininities. In an era of individualism and post-critique, this is not easy, which makes it even more urgent to reinforce the training of future teachers in gender issues and to make them used to analyzing discourses and practices from a feminist standpoint.

Within this new educational paradigm, focused on competences rather than contents and aiming at social justice and equality, research around the hidden discourses conveyed at schools, that is, the hidden curriculum [22], becomes paramount. Soto-Molina and Méndez [50] list the issues that teachers of English may face from an intercultural point of view: dilemmas, contradictions, and challenges. Paraphrasing them from a gender perspective, in ELT environments we are faced with:

- The dilemmas to overcome the imbalance between men and women;
- The contradictions between our resources, materials, and contexts and our teaching discourses and practices;
- The challenges to adopting critical pedagogies to work with gender approaches.

Teachers-to-be and in-service teachers need to be trained to face these issues critically, which necessarily involves knowing about femininities and masculinities and their interactions, to be able to discover the (not so) hidden messages presented before. However, the global neoliberal context that has permeated education does not enhance it. In this view, education has become an apparently neutral commodity and a service rather than a right. As a result, education offers a wide array of services which can be freely chosen as long they can be purchased [51,52]. Probably, ELT is one of those educational services that better epitomizes this logic: supposedly a neutral global language [53], it is "purchased" by parents who legitimately want their children to do well in life. Instead of a tool to communicate, the language then becomes one more means to perpetuate linguistic imperialism, available only to a few [50,54].

Applying this neoliberal mindset to the content of the education "services", are all alternatives offered or just a few? What visions about "doing woman" or "doing man" are communicated? In post-critique times, are they questioned? Are they hierarchically organized? Are any hostile sexist discourses conveyed? What about benevolent sexist messages? To open ELT to this kind of questions would help to humanize it, to acknowledge the "life capital" involved—the people who enact it and generate "new vistas" around it:

> This means that rather than describing our phenomena and participants through discourses that succumb to a dualism of "good" and "bad", "motivated" or "demotivated" or "having or not having a certain skill" [or "man" or "woman"], we take an approach that paints a fuller picture, and which confers respect to the human stories. [55] (p. 1402)

Resorting to these issues—together with others—would contribute to contesting the neoliberal banking education model based on the accumulation of knowledge [56]. Moreover, it would also aim at educating critical citizens, able to see the social reality beyond binaries and aware of unfair social relationships. This would certainly be a more engaged pedagogy [54]. What is more, just from the perspective of ELT, it could be a very effective way to trigger discussion and situated real communication to motivate students.

## 7. Conclusions

The aim of this article was to theoretically support the validity of femininities as relevant variables in ELT training, practices, and, above all, research. Focusing on the last one, it has been labeled as "feminist academic activism" through the application of Feminist Critical Discourse Analysis [10,11]. With this aim being achieved, the ideas posed here ought to be put into practice through empirical studies, which will be the logical continuation of this work.

But, what areas of ELT should be the object of such analysis? To answer this question, in a previous work of mine focused on masculinities [1], I drew inspiration from Roiha and Polso [57] and their *5-Dimensional Model* to differentiate and accommodate teaching in Finland, aimed at learners with special needs. I then adapted them to the teaching of English, conceived as another tongue [2], as follows, now listing the suggested main areas of study in ELT when researching femininities, adding some possible examples:

- Teaching arrangements: grouping, co-teaching, language assistants, support teachers: For example, can the decisions underlying groupings and roles be explained as the enactment of concepts presented before, such as benevolent sexism [8]? Do groups interact differently with one another when they are mostly boys or girls? Similarly, do groups interact differently with feminine or masculine teachers?
- Learning environments: physical and social characteristics: For example, who tends to participate more, boys or girls? How are femininities and masculinities performed by educational subjects, such as learners and teachers? What femininities are privileged in class and thus given a stronger voice? How do teachers relate with learners who embody "pariah" femininities [37], such as lesbians and strong female learners? How about their classmates?
- Teaching methods, projects, language skills (listening, speaking, reading, and writing), and intercultural and citizenship competences: For example, do female teachers tend to opt for different teaching methods? How can they be compared to those chosen by male teachers? Even more importantly, do teachers include critical activities that promote questioning inequalities beyond the purely hostile realization of sexism, thus including benevolent sexism [8]? Do teaching techniques promote questioning gender itself? Are social-justice-related issues discussed in class, helping understand the interaction between different "axes of difference" [19]?
- Support materials, ICT, songs, stories, and course books: For example, do groups formed by mostly girls or boys prefer different support materials? Do boys and girls engage differently with different resources? Are all femininities represented and included in lessons, or just the "emphasized" ones [4]? Are there instances of "pariah femininities" [37]? Do contents and resources privilege instances of entitled femininities [46]?
- Assessment, initial, formative, summative, and tools (rubrics, checklists, exams, quizzes, presentations, journals, and portfolios): For example, do girls prefer different assessment tools from those chosen by boys? Do girls consistently do better or worse in a certain type of assessment than boys? Does assessment promote uncritical repetition of contents, or does it facilitate critical thinking to develop social and citizenship competences too?

Being familiar with the ideas presented in this article would help explain these gender differences in ELT through the discourses around femininities that are perpetuated in this teaching practice. As this list evidences, it is impossible to empirically research on femininities without considering masculinities, which is the next step to come in this study. Still, when this further stage comes into effect, it will be important not to analyze them as contradictory binaries, but as just two possible gender identities in a rich continuum that also includes, for instance, transgender, non-binary, and gender-fluid individuals, who should therefore be added to the questions above.

In short, it is the acknowledgment of intersectionality in educational environments: "A metaphor for understanding the ways that multiple forms of inequality or disadvantage

sometimes compound themselves and create obstacles that often are not understood among conventional ways of thinking" [58] (p. 149). In other words, the fact that this project focuses on certain aspects of the hidden *gender* curriculum that can be discovered in ELT contexts should not make us forget that the limitations imposed by femininities and masculinities overlap with those carried out by other social institutions like ethnicity, nationality, nativespeakerism, sexual orientation, wealth, or religion, among others that also interact with gender in ELT classrooms. Dealing with them all in a single article would burden it and, probably, distract the reader from the text's focus, which is why intersectionality is acknowledged here, but not particularly dealt with.

After all these pages, it becomes clear that femininity and all its different realizations, being no more than social institutions [5], cannot be regarded as appropriate descriptive terms to apply to those involved in ELT. Rather, they are "categories [that] constitute unlivable constraint" [59] (p. 8). Still, they should be embraced as insightful variables to get further understanding about some aspects of the hidden gender curriculum [60] present in ELT classes, that is, the identitarian discourses around men and women that actively populate this educational field. The committed application of such feminist critical discourse analysis, looking into the interaction of masculinities and femininities with other "axes of difference" [19], would certainly help to transform the teaching of English—"the map of our failures"—at least from a gender perspective, so that it is no longer "the oppressor's language [that] yet I need it to talk to you" [61] (p. 117).

**Funding:** This research received no external funding.

**Institutional Review Board Statement:** Not applicable.

**Informed Consent Statement:** Not applicable.

**Data Availability Statement:** No new data were created or analyzed in this study. Data sharing is not applicable to this article.

**Conflicts of Interest:** The author declares no conflict of interest.

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
