# Peer review of "Feminist Academic Activism in English Language Teaching: The Need to Study Discourses on Femininities Critically"

_education, doi:10.3390/educsci13060616_

Round 1

Reviewer 1 Report

When I received the manuscript I was genuinely excited because I am a feminist and an English Language Teaching expert. Clearly, my expectations were high, which is possibly the reason that I feel the paper falls short of delivering what it purported to deliver. For example, as it says in the abstract, the paper "aims to theoretically support the assumption of Feminist Critical Discourse Analysis as a suitable method to discover how all these social phenomena interact in ELT contexts, helping to shape its (gender) hidden curriculum." and "it offers areas of interest for critical researchers and ELT practitioners". As an ELT researcher and practitioner, I find this paper does not, unfortunately, achieve either. 

The paper provides a fantastic overview of issues and concepts around feminisim, yet it does not have the least bit to do with English Language Teaching. Thus, the title, the abstract, and the aim in the introduction are misleading, even the first heading "Construction of gendered identities in ELT".

There is a wealth of sources on a number of issues in ELT, from stereotypes in coursebooks (e.g. representation of female characters), native speakerism, pay gap between NESTs and NNESTs, to decentering whiteness in the industry of English Language Teaching (if the author wants to delve into intersectionality). The author casually gives the names of the authors of several studies on the role of gender in ELT (lines 107-109 and again 345-347) but does not devote a single line to the findings of these studies, which would be much more beneficial. Providing an overview of where ELT is in terms of gender stereotypes through an exploration of current studies on the matter would be an essential part of an article that claims to explore femininities in ELT. It Too much space in the article is taken up by elaborating on well-established notions of gender and feminism, while no space is given to the world of ELT and its numerous issues as they relate to women as students, teachers, authors, trainers, etc. This disbalance can easily be seen in the list of references.

While I understand that the paper is one in a series, it is simply too abstract and not even tangentially related to ELT; an article needs to be able to stand on its own as a piece of scholarship, not to be interpreted only in relation to its prequels and sequels. 

To conclude, the authors have provided a thorough and clear overview of feminism and related concepts, but have not successfully connected it to English Language Teaching in any way. For this reason, I find the article needs to be either thoroughly rewritten or reframed. 

To make the language more formal, please do not use contracted forms, for example, "hasn't" should be "has not", etc.

Author Response

Dear reviewer,

First of all, I’d like to thank you for the time and effort you have put into reading and assessing my work.

I understand you may have expected more practical guidance in this paper, but it is an argumentative, narrative literature review, not an empirical study. I am sorry I have disappointed you: I never intended to explore femininities in ELT (I will soon), just to support its need, the adoption of “femininities” (not just gender) as an analytical category, never as a descriptive/identitarian one.

To avoid disappointing prospective readers too, I have changed the title, which I think looks less empirical now. I have also emphasized the fact that it is a theoretical work, with the same intention. I do think I deal with ELT, as a context for my arguments, with the works I cite. I could say more about them, but I feel that would probably be deviating from the article’s main idea, and could end up confusing the reader (more). I think it is acquaintance with the ideas around femininities that practitioners lack, and this is essential to later find them in resources and lessons. Let me insist, that will be the next stage. Still, I have tried to make better use of those sources and shortly listed some of the biases they uncover.

I think the article is able to stand on its own, just as “theory”, of course. But a future article that empirically studies femininities and masculinities wouldn’t be able to accommodate so many ideas that can later be identified in the findings. This is why I chose to write a series of three: the one on masculinities is already online, and the empirical work will soon follow, once this second article has (hopefully) been published.

Thank you for suggesting deleting the negative contractions. I have done it, and it reads a lot better.

I hope to have been able to —at least slightly— reframe the text so that now it is easier to understand and follow.

I apologize again for the disappointment, and honestly thank you for your review, which has helped me much.

Best,

Author

Reviewer 2 Report

1. Relevant issue addressed, theme and originality, bringing some novelty in relation to the material in the specific area.  

2. The work is original and representative of the scope of the journal. However, some changes should be made: a Method section should be included, describing the methodology used, as well as a list of reference authors used.

3. The conlusions are consistent with the research carried out and the bibliographical references are adequate.

The research can be published with the minor changes suggested above.

Author Response

Dear reviewer,

First of all, I’d like to thank you for the time and effort you have put into reading and assessing my work.

To address your suggestion, in the abstract it is now clear that it is an argumentative, narrative review. In addition, in the introduction, I have added a paragraph to explain what that kind of review is, and listed the most important sources included in the references.

I hope these changes sufficiently address the issue you highlighted. I have, of course, made some other changes according to the other reviewers’ remarks. I hope you will enjoy text better now.

I can’t finish this answer without thanking you immensely for the favorable recommendation you have provided so far and, particularly, for being willing to sign your review.

Best,

Author

Reviewer 3 Report

I think this is a wonderful paper and an important contribution to the field. I think the one thing that is missing is a description of the intersectionality of gender identity and racial/ethnic identity such as more broadly seen in the work of Kimberle Crenshaw and more specifically in the work of bell hooks and other more contemporary authors. Especially since the work is centered around English Language Teaching and gender I think it is a missed opportunity to not dig into the how the hegemony of English is especially dismissive of many group's racial and ethnic identities in which English has been foist upon these groups as a method of participating in larger society instead of honoring cultural languages, especially for people from Indigenous and First Nations communities and the intersection of this with gender.  I think if you don't want to do that, at least please acknowledge that more directly as a limitation of this paper.

Additionally, I think more specific strategies for surfacing and honoring the identities of the students and educators in the room would be a welcome addition to make this both a theoretical and practical piece of work.

Author Response

Dear reviewer,

First of all, I’d like to thank you for the time and effort you have put into reading and assessing my work.

I couldn’t agree more on your arguments, and I have tried to give more relevance to intersectionality. For sure, not enough, you are right about that; but I have the feeling including more issues could “distract” the reader from the main ideas around femininities, and make the article more “obscure”. I know it isn’t an easy text, and I don’t think including too much would work. Still, better mention of intersectional issues is made now, and the lack of more serious resort to it is acknowledged.

I hope you will find the changes applied beneficial for the general understanding of the text and, especially, that you will enjoy it better.

Finally, I wouldn’t dare finish this answer without thanking you immensely for the complimentary words you used to introduce your review, as well as for willing to sign your review. Thank you so, so much!!!!!

Best,

Author

Round 2

Reviewer 1 Report

Congratulations on a much better version of the article!